# A Combined Approach to the Prevention of Postoperative Atrial Fibrillation in Cardiac Surgery

**DOI:** 10.3390/biomedicines13081999

**Published:** 2025-08-17

**Authors:** Mariia L. Diakova, Mikhail S. Kuznetsov, Yuri Yu. Vechersky, Elena B. Kim, Stepan V. Zyryanov, Konstantin A. Petlin, Boris N. Kozlov

**Affiliations:** Cardiology Research Institute, Tomsk National Research Medical Center, Russian Academy of Sciences, 634012 Tomsk, Russia; kms@cardio-tomsk.ru (M.S.K.); vjj@cardio-tomsk.ru (Y.Y.V.); ekim@cardio-tomsk.ru (E.B.K.); zyryanzone@gmail.com (S.V.Z.); kpetlin@yandex.ru (K.A.P.); bnkozlov@yandex.ru (B.N.K.)

**Keywords:** colchicine, cardiac surgery, cardiopulmonary bypass, postoperative atrial fibrillation, pericardial fenestration, pericardial drainage

## Abstract

**Background:** Postoperative atrial fibrillation (POAF) is a common complication after cardiac surgery with cardiopulmonary bypass (CPB) affecting between 5% and 40% of patients, which leads to hemodynamic instability, an increased risk of thromboembolism, decompensated heart failure, prolonged hospitalization, and higher treatment costs. Currently, there are no universally accepted guidelines for preventing POAF. **Methods:** A single-center, prospective, randomized controlled trial, “The Effect of Colchicine on the Occurrence of Atrial Fibrillation after Cardiac Surgery” (CAFE), ClinicalTrials.gov ID: NCT06798714, was conducted. The study included 140 patients with coronary artery disease randomized into two groups of 70 patients each. Group 1 (control group) received standard postoperative care. Group 2 (intervention group) received colchicine (Colchicum-Dispert at a dose of 500 mcg 4 h before coronary artery bypass grafting (CABG) with CPB and at a dose of 500 mcg twice daily for 10 days postoperatively) and underwent intraoperative pericardial fenestration using an original technique. **Results:** Perioperative colchicine administration combined with intraoperative pericardial fenestration reduced POAF incidence to 2.9% compared to the control group with POAF incidence of 12.9% (*p* < 0.05). This management strategy was not associated with an increased incidence of infectious complications, gastrointestinal disorders, or elevated levels of alanine aminotransferase, aspartate aminotransferase, or creatinine. **Conclusions:** Perioperative colchicine administration combined with pericardial fenestration during CABG with CPB is associated with a reduced POAF incidence, good tolerability, and does not contribute to an increased incidence of infectious complications or impaired liver and renal function.

## 1. Introduction

Atrial fibrillation is reported to occur in 16% to 50% of patients after cardiac surgery [1,2]. Postoperative atrial fibrillation (POAF) significantly contributes to an increased risk of mortality and associated complications [3].

It is suggested that activation of the systemic inflammatory response (SIRS) plays a significant role in the pathogenesis of postoperative atrial fibrillation (POAF), which inevitably develops following exposure to cardiopulmonary bypass (CPB) during cardiac surgery. SIRS, which is generalized and systemic in nature, is based on the activation and release of various cellular, extracellular, and organ mediators [4]. In addition to SIRS, the following local factors related to surgery also play a significant role in the pathogenesis of POAF during cardiac surgery: atrial trauma, surgery-related pericardial inflammation, increased atrial pressure (including due to left ventricular diastolic dysfunction), autonomic nervous system dysfunction, metabolic abnormalities and electrolyte imbalance, or ischemic myocardial injury occurring during surgery [5,6].

Therefore, considering complex pathogenesis, the prevention of POAF requires addressing both systemic and local inflammation following cardiac surgery.

Currently, there are no universally accepted approaches for preventing SIRS. In practice, nonsteroidal anti-inflammatory drugs (NSAIDs) are most commonly used, while glucocorticosteroids are used less frequently. However, the use of NSAIDs and glucocorticosteroids has its contraindications and numerous possible side effects, which limit their use in cardiac surgery patients. The feasibility of using colchicine for reducing SIRS and the incidence of POAF development in cardiac surgery patients is being widely discussed [7,8,9].

Colchicine is a potent anti-inflammatory agent. Its mechanism of action involves reducing leukocyte migration to the site of inflammation, an antimitotic effect through the suppression (complete or partial) of cell division during anaphase and metaphase, and the prevention of neutrophil degranulation [10].

To reduce local cardiac inflammation after surgery, adequate drainage of the pericardial space is needed for complete and timely evacuation of inflammatory exudate. The procedure of posterior pericardiotomy is known to allow for pericardial drainage. The technique of posterior pericardiotomy involves making a linear incision to drain the pericardial space. The disadvantage of this method is the risk of premature “closure” of the pericardium and cessation of adequate drainage of the pericardial space. In a systematic review and meta-analysis [11], posterior pericardiotomy was associated with effective prevention of pericardial effusion, cardiac tamponade, and a new onset of POAF following coronary artery bypass grafting (CABG) in adults, as well as with a small number of complications. Such outcomes indicate that posterior pericardiotomy is a simple and safe surgical technique without obvious complications. However, this meta-analysis presented a number of limitations that preclude its findings from being considered definitive [11]. Thus, more high-quality randomized controlled trials (RCTs) are still needed to assess the safety of posterior pericardiotomy for the prevention of POAF following CABG.

In addition to the posterior pericardiotomy technique, there is a method for the prevention of cardiac tamponade after cardiac surgery as well [12], which allows for adequate prevention of cardiac tamponade after cardiac surgery throughout the postoperative period, providing adequate pericardial drainage and timely removal of exudate in the postoperative period, helping to reduce local inflammation and, accordingly, reduce the risk of paroxysmal atrial fibrillation.

Given all of the above, we opted to propose a novel approach to prevent POAF, which combines two distinct interventional strategies (pharmacological and cardiac surgical) designed to reduce both local and systemic inflammation. The lack of prior research in this area was the primary motivation for the current study.

The aim of our study was to investigate the clinical effectiveness of combining two approaches—pharmacological (administration of colchicine to reduce SIRS) and surgical (performing pericardial fenestration and drainage during surgery to reduce local inflammation)—as a method for preventing POAF in cardiac surgery patients undergoing CABG with CPB.

## 2. Materials and Methods

A single-center, prospective, randomized controlled trial, “The Effect of Colchicine on the Occurrence of Atrial Fibrillation After Cardiac Surgery” (CAFE), registered at ClinicalTrials.gov ID: NCT06798714, was conducted at the cardiac surgery department 1 and the department of anesthesiology and intensive care of Cardiology Research Institute in Tomsk. The study protocol was approved by the local ethics committee of Cardiology Research Institute, protocol No. 245 dated 28 June 2023. The study was conducted in accordance with the principles of the Declaration of Helsinki and ethical guidelines for epidemiological studies of the Government of the Russian Federation.

The inclusion criteria for the study were: patients with stable coronary artery disease (CAD), with multivessel coronary artery disease confirmed by coronary angiography, scheduled for CABG with CPB, and who signed informed consent to participate in the study.

To exclude the possible influence of any other factors affecting the results of the study and to create a homogeneous group of patients, exclusion criteria were developed as follows: reduced left ventricular ejection fraction (≤35%) according to echocardiography, heart valve disease requiring surgical treatment, hepatic insufficiency with elevated levels of liver transaminases (alanine aminotransferase (ALT) and aspartate aminotransferase (AST)) ≥ 1.5 times the upper limit of normal, renal insufficiency (reduced glomerular filtration rate (GFR) less than 35 mL/min/1.73 m^2^ according to the CKD-EPI equation), a history of permanent, paroxysmal, or persistent atrial fibrillation, a history of hypersensitivity, neutropenia, or alcoholism. Refusal to sign the informed consent form was also an exclusion criterion.

For the study, 140 patients were randomized into two groups using random allocation (Figure 1). Randomization was conducted via numbered opaque envelopes. The sequential numbers on the envelopes served as randomization identifiers, recorded in the patient’s Individual Registration Card. Personnel involved in outcome interpretation were blinded to the assigned therapy until study completion. Group 1 comprised 70 patients who received standard treatment, including the administration of NSAIDs—diclofenac at a daily dose of 100 mg/day for 5–7 days postoperation. Additionally, posterior pericardiotomy and isolated pericardial drainage were performed intraoperatively, following the generally accepted standard technique. In Group 2, all patients (*n* = 70) received colchicine (Colchicum-Dispert, manufactured by HAUPT PHARMA WULFING, GmbH (Gronau (Leine), Germany), registration certificate holder: PHARMASELECT INTERNATIONAL BETEILIGUNGS, GmbH (Hannover, Germany)) at a dose of 500 mcg once 4 h before surgery, followed by 500 mcg twice daily for 10 days postoperation. No other anti-inflammatory drugs were administered. Furthermore, patients in Group 2 underwent pericardial fenestration and drainage intraoperatively using an original technique [12].

**Surgical Technique for Pericardial Fenestration and Drainage**. In the operating room, with the patient under combined anesthesia and mechanical ventilation, the surgical site is prepped with an antiseptic solution and draped. The skin and subcutaneous tissue are incised, and a median sternotomy is performed. A wide T-shaped pericardiotomy is carried out. The method of CPB connection is chosen based on the planned extent of the surgery. The main stage of the cardiac surgery is then performed. Before weaning from CPB, the apex of the heart is retracted laterally, and a pericardial fenestration is created in the left mediastinal portion, measuring up to 4 cm in diameter, below the phrenic nerve. CPB is then discontinued. At the completion of the surgery, a drain is placed in the diaphragmatic region of the pericardium and advanced into the left mediastinal area of the pericardium. This drain is passed through the previously created fenestration and then positioned within the left pleural sinus, ensuring that the perforated section of the drain remains entirely within the sinus. The retrosternal space is drained separately with another drain. The sternum is closed using wire cerclage. The wound is closed in layers. An aseptic dressing is applied. The pericardial drainage technique is shown in Figure 2A,B.

CABG was performed using a standard technique, with circulatory arrest, under CPB and cold cardioplegia with Custodiol solution. Autologous arterial grafts—arteria mammaria sinistrae—and autologous venous grafts—vena saphena magna—were used as conduits. When the arteria mammaria sinistrae was not suitable for use as a graft, complete autologous venous grafting was performed. In the early postoperative period, patients were admitted to the intensive care unit (ICU), where bedside monitoring of electrocardiograms, arterial blood pressure parameters, respiration, and thermometry was conducted. Monitoring of key biochemical blood parameters, electrolytes, acid-base balance, and hemostasis was also performed. If the hemoglobin level decreased below 80 g/L in the complete blood count, red blood cell transfusions were administered. After recovery and stabilization of essential vital functions, the patient was transferred to a general ward, where they were further monitored by the attending cardiologist and the operating cardiac surgeon until the time of discharge. Postoperatively, patients in both groups received beta-blockers, in the absence of contraindications, bradycardia (heart rate < 60 bpm), or intracardiac conduction abnormalities.

The primary endpoint of the study was the incidence of POAF, which was determined by the presence of a registered arrhythmia episode with typical ECG findings, with a minimum duration of 30 s. Atrial fibrillation during the patient’s stay in the ICU after surgery was diagnosed based on bedside monitoring data. After transfer from the ICU to the general ward, diagnosis was based on data from daily scheduled electrocardiogram recordings, as well as the occurrence of complaints regarding heart rhythm disorders throughout the hospitalization period, or the identification of rhythm disorders during daily examination by the attending cardiologist.

Secondary endpoints included the incidence of infectious complications in the postoperative period, specifically pneumonia and mediastinitis; as well as markers of systemic inflammation: white blood cell count in the complete blood count, C-reactive protein (CRP) levels in the blood (mg/L) (measured preoperatively, and on days 1, 3, 5, and 10 postoperatively). AST (U/L), ALT (U/L), and creatinine (µmol/L) levels were also monitored preoperatively, and on days 1, 5, and 10 postoperatively, as a screening for the development of kidney and renal dysfunction. Throughout the postoperative period until discharge, the incidence of gastrointestinal disturbances (nausea, diarrhea, vomiting) was assessed based on patient complaints. Using ultrasound examinations performed daily postoperatively until discharge, the amount of fluid accumulation in the pericardium and pleural cavities was quantitatively assessed. The need for and number of pleural taps due to fluid accumulation in the pleural cavities (more than 300 mL by ultrasound) were also monitored. Operative mortality and the occurrence of major postoperative adverse events, including stroke, myocardial infarction, and postoperative bleeding, were analyzed as well.

Statistical analysis was performed using STATISTICA 12.0 (StatSoft, Inc., 1984–2011, Tulsa, OK, USA). Quantitative data were presented as median and quartiles, Me [25th; 75th percentile]. Comparisons of quantitative data between groups were performed using the Mann-Whitney U test. For comparisons of qualitative data, Fisher’s exact test was used. Odds ratios (ORs) and 95% confidence intervals (CIs) were calculated using the online StatTech program (https://medstatistic.ru/calculators/calcrisk.html (accessed on 4 June 2025)). The numbers needed to treat and harm (NNT/NNH) were calculated using the online ClinCalc tool at https://clincalc.com/ (accessed on 5 June 2025). Differences were considered statistically significant at *p* < 0.05.

## 3. Results

The clinical and demographic data as well as CPB time and the number of anastomoses performed during surgery, are presented in Table 1.

Postoperative data are presented in Table 2**.**

When analyzing the postoperative data, the incidence of POAF was significantly higher in Group 1, occurring in 12.9% (9 patients) compared to 2.9% (2 patients) in Group 2 (NNT/NNH was 10.0, odds ratio OR—0.199, 95% CI: 0.041; 0.959, *p* = 0.03). The majority of POAF occurred on days 1–4 postoperatively—in eight patients in Group 1, and in one patient on day 7. In Group 2, POAF developed on day 3 in one patient, and episodes were recorded on days 3 and 5 postoperatively in another patient. In all cases, the POAF episodes were terminated with intravenous amiodarone.

There were no deaths in either group. The rates of other major adverse events (stroke, myocardial infarction, and postoperative bleeding) were comparable between groups.

When analyzing the changes in CRP levels during the postoperative period, a significant increase in blood CRP was observed from days 1 to 10 after surgery compared to preoperative data in both groups. The CRP level was higher in Group 1 on postoperative day 3, but lower on postoperative day 10, compared to patients in Group 2.

WBC count in the postoperative period was higher in the group of patients receiving colchicine on the next day after surgery compared to patients in Group 1. Subsequently, the WBC count in blood tests was comparable between patients in both groups.

No significant between-group difference was found in AST, ALT, and creatinine levels during the postoperative period.

The incidence of mediastinitis, pneumonia, postoperative wound infections, requiring surgical debridement of the sternum, as well as the incidence of hydrothorax requiring thoracentesis between the groups, was comparable.

In group 1, one patient complained of nausea and one episode of vomiting on the first day after surgery, while AST and ALT levels did not increase. In group 2, one patient experienced an exacerbation of pancreatitis on day 7 of the postoperative period, which was terminated with conservative therapy.

## 4. Discussion

Current understanding suggests that the development of POAF is multifactorial and attributable to the combined action of various triggers and substrates [13,14], initiating the progression of atrial cardiomyopathy. Atrial cardiomyopathy is defined as any complex of structural, architectural, contractile, or electrophysiological alterations involving the atria that can result in clinically significant manifestations [15]. This definition is also applicable to the development of POAF.

POAF following cardiac surgery continues to be a significant complication. Research is ongoing to prevent the occurrence of this complication, including investigations into the efficacy of colchicine therapy; however, study findings are inconsistent. For instance, while the COPPS and COCS sub-studies demonstrated high efficacy of colchicine for POAF prevention, the COPPS-2 trial, conversely, reported limited efficacy of the drug [16,17,18,19].

Meta-analyses of RCTs have demonstrated a correlation between colchicine use and a significant decrease in POAF incidence [6,20]. Nevertheless, the optimal dose and duration of colchicine for preventing POAF remain undetermined, requiring additional studies.

Deficiencies in the conducted clinical studies include small and heterogeneous patient cohorts, including those with valvular and non-valvular pathologies undergoing various interventions such as CABG, valve replacement, or concomitant cardiac surgery procedures (COCS, COPPS studies). Valvular pathology is known to be linked to impaired intracardiac hemodynamics and altered cardiac chamber sizes, inherently increasing arrhythmia risk. The END-AF Trial reported a lower POAF incidence (19.2%) in patients undergoing isolated CABG compared to those with concomitant cardiac surgeries (23%) [21]. Furthermore, colchicine therapy was more effective in the isolated CABG group (13.2% POAF incidence) than in the concomitant cardiac surgery group (18.4% POAF incidence). For example, Zarpelon et al. included patients without a valvular disease who underwent CABG with CPB [22].

Our ”CAFÉ” study utilizes a single methodology that encompasses colchicine administration alongside pericardial fenestration and integrated drainage of the pericardium and left pleural cavity through a single drain. This study included a homogeneous group of patients with CAD, excluding those with significant valvular pathology or severe chronic heart failure. All participants underwent CABG with CPB and no valve procedures, ensuring comparable baseline intracardiac and systemic hemodynamic profiles. Existing RCTs investigating the efficacy of colchicine for preventing POAF lack data regarding the performance of pericardial drainage during surgical intervention. However, in our view, this data is highly significant, as adequate pericardial drainage in the early postoperative period undoubtedly plays a crucial role in reducing the incidence of POAF by facilitating timely exudate drainage and mitigating local inflammation.

Pericardial drainage achieved through posterior pericardiotomy has been widely adopted since the early 2000s. This technique is easily reproducible and requires no specific skills or specialized instrumentation. Limitations to its application, similar to those for pericardial fenestration and drainage using our original technique [12], include the presence of extensive adhesions in the left pleural cavity, which can technically preclude the procedure. In published studies on the use of posterior pericardiotomy as a method for POAF prophylaxis, the emphasis is primarily on the technical aspects of the maneuver, with little attention given to concomitant anti-inflammatory therapy. For instance, ”The Effect of Posterior Pericardiotomy on the Incidence of Atrial Fibrillation After Cardiac Surgery” (PALACS) study [23] only reports data on beta-blocker administration. Furthermore, the patient cohort included in that study was heterogeneous, comprising patients requiring CABG, aortic, and aortic valve interventions. It is also not entirely clear how pericardial drainage was carried out in their control group. The incidence of POAF in the group with posterior pericardiotomy was 17%. A retrospective analysis comparing posterior pericardiotomy with isolated pericardial drainage in patients undergoing isolated CABG demonstrated the efficacy of this approach in reducing POAF to 20.2% versus 26.3% (*p* < 0.05) [24]. However, this study also lacked data on concurrent anti-inflammatory therapy, and the included patient groups were not homogeneous.

Our “CAFÉ” study reported a POAF incidence of 12.9% in the control group, which is comparable to the findings of a similar study by Zarpelon et al. [22]. However, our POAF prophylaxis approach proved more efficacy, as the incidence of POAF in the group treated with colchicine, fenestration, and pericardial drainage was 2.9%, compared to 7.04% reported by Zarpelon et al. [22] for colchicine therapy.

Therefore, our combined approach of administering colchicine and performing pericardial fenestration according to the original methodology [12] proves effectiveness in preventing POAF development.

Cardiac surgery performed under CPB is known to activate the systemic inflammatory response syndrome. In addition to systemic inflammation induced by CPB, one must also consider the local inflammation developing in the pericardium following direct surgical manipulations of the heart. In response to damage to pericardial mesothelial cells, local inflammation—pericarditis—develops, which is then exacerbated by NLRP3 inflammasome activation [25,26]. The role of NLRP3 inflammasome activation in the development of pericarditis has been confirmed by various researchers [27]. Furthermore, studies demonstrate that the NLRP3 inflammasome directly contributes to POAF development and is upregulated in patients with prevalent forms of atrial fibrillation, including paroxysmal, persistent, and POAF [28,29,30]. Colchicine has the ability to disrupt NLRP3 inflammasome activation [25], which also provides a rationale for prescribing this drug to patients who have undergone cardiac surgery.

Our study did not evaluate the NLRP3 inflammasome; however, it analyzed the changes in CRP and leukocyte levels, serving as markers of postoperative inflammation severity. We observed higher CRP levels in Group 2 on postoperative days 1 and 3 compared to Group 1. Furthermore, WBC counts were slightly higher in Group 2 a day after surgery. These findings might be attributed to more traumatic pericardial injury during drain insertion. However, on postoperative day 10, CRP levels were lower in Group 2, and WBC counts were comparable in both groups, suggesting a pronounced anti-inflammatory effect of colchicine in patients who underwent CABG.

While colchicine’s mechanism of action, which suppresses the inflammatory response, might lead one to expect an increase in infectious complications, our study found that colchicine treatment was not associated with a higher rate of postoperative infectious complications. This aligns with similar results observed in a meta-analysis of other RCTs [7].

Colchicine is known to potentially cause hepatic and renal failure. In the context of cardiac surgery performed under CPB, the onset of such complications would significantly compromise the postoperative course and prolong hospital admission. Thus, we deemed it important to monitor for liver and kidney dysfunction in patients receiving this therapy. Our analysis of postoperative ALT, AST, and creatinine levels, however, demonstrated no disparities between the standard therapy group and the colchicine therapy group. This indicates that colchicine is safe when administered in small doses for a short 10-day duration postoperatively.

Data from prior RCTs indicate that gastrointestinal disturbances (nausea, diarrhea, vomiting) are the most common adverse events associated with colchicine therapy [2]. The colchicine treatment protocol implemented in our study did not demonstrate an association with a higher incidence of these complications.

Our study included patients with similar underlying pathology and undergoing the same type of cardiac surgery to exclude the potential influence of confounding factors that trigger POAF. Specifically, patients with severe valvular pathology, which could induce intracardiac hemodynamic disturbances and precipitate cardiac rhythm disorders, were excluded. This controlled study design highlighted the high effectiveness of the applied methodology in the selected patient population.

Limitations of the Study. This is a single-center study conducted in patients undergoing isolated CABG. The methodology involving pericardial fenestration and integrated drainage of the pericardium and left pleural cavity via a single drain is well-practiced and readily performed by the cardiac surgeons at our institution. Whether this technique can achieve comparable outcomes in patients requiring different cardiac surgical procedures (e.g., concomitant CABG and valve repair/replacement) or in patients of other centers remains to be determined and warrants further research. Furthermore, a relatively small sample size constitutes an additional limitation of our study.

## 5. Conclusions

The implemented approach in our study demonstrated high effectiveness in preventing POAF in patients with CAD undergoing CABG under CPB. A combination of systemic anti-inflammatory therapy with low-dose colchicine (500 mcg administered 4 h prior to surgery and 500 mcg twice daily for a short 10-day course), coupled with the surgical performance of pericardial fenestration and drainage, resulted in a reduced POAF incidence of 2.9%. Notably, this combined strategy was not associated with an increased risk of postoperative complications, including infectious issues (pneumonia, mediastinitis, sternal wound infections), gastrointestinal disturbances, or compromised liver and kidney function.

## Figures and Tables

**Figure 1 biomedicines-13-01999-f001:**
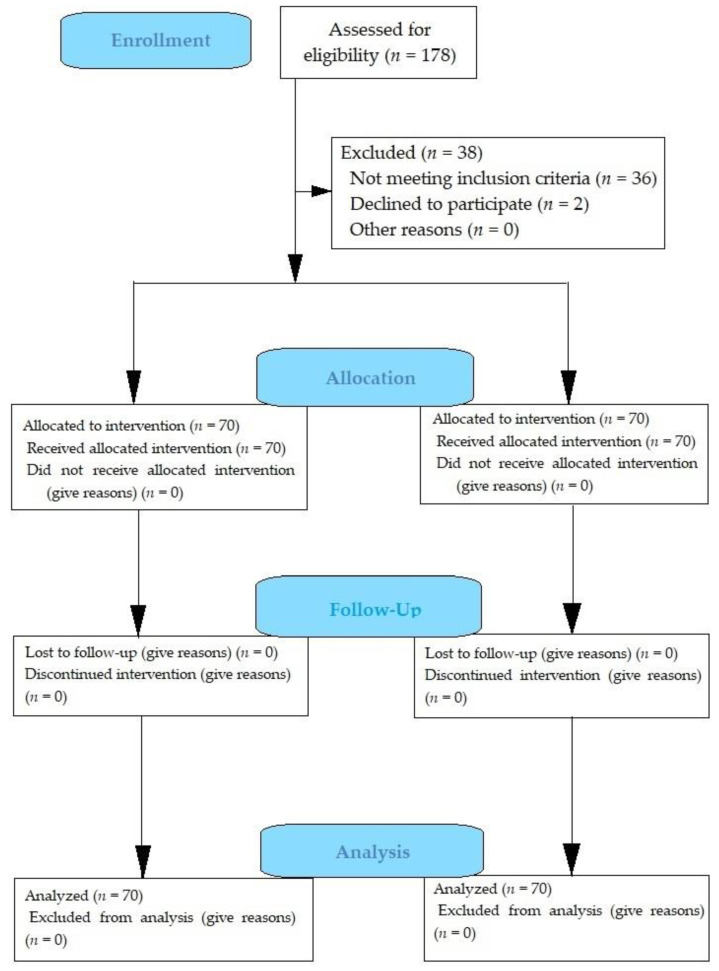
Flowchart of the study design.

**Figure 2 biomedicines-13-01999-f002:**
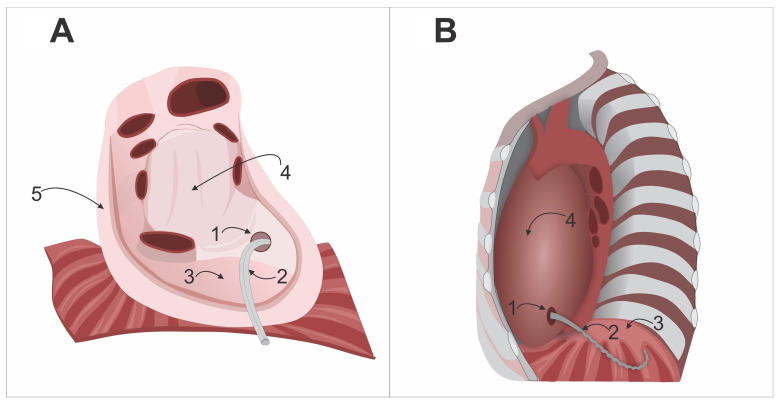
Schematic illustration of pericardial fenestration and drainage after cardiac surgery. (**A**)—anterior wall of the pericardium removed, heart removed at the entry and exit of major vessels: 1—pericardial fenestration, 2—drain, 3—diaphragmatic surface of the pericardium, 4—left mediastinal portion of the pericardium, 5—pericardium; (**B**)—view from the left pleural cavity (left lung removed): 1—pericardial fenestration, 2—drain, 3—left pleural sinus, 4—left mediastinal portion of the pericardium.

**Table 1 biomedicines-13-01999-t001:** Clinical and demographic data.

Variables	Group 1 (*n* = 70)	Group 2 (*n* = 70)	*p*-Value
Age, years (Me [25; 75])	62.3 [57; 69]	62.5 [57; 67]	0.40
Gender: male (*n*, %) female (*n*, %)	58 (82.9) 12 (17.1%)	59 (84.3%) 11 (15.7%)	0.82
CCS class 2 (*n*, %) CCS class 3 (*n*, %) Silent myocardial ischemia (*n*, %)	17 (24.3%) 49 (70%) 4 (5.7%)	24 (34.3%) 45 (64.3%) 1 (1.4%)	0.39 0.47 0.10
History of MI (*n*, %)	40 (57.1%)	39 (55.7%)	0.86
Hypertension (*n*, %)	66 (95%)	67 (95%)	0.70
Type 2 diabetes mellitus (*n*, %)	19 (27%)	25 (35.7%)	0.27
BMI (kg/m^2^) (Me [25; 75])	29.2 [25.5; 32.0]	28.7 [26; 30.8]	0.64
Normal weight (*n*, %) Overweight (*n*, %) Obesity class 1 (*n*, %) Class 2 (*n*, %) Class 3 (*n*, %)	18 (25.7) 26 (37.1%) 19 (27.1%) 4 (5.7%) 3 (4,3%)	18 (25.7) 31 (44.3%) 18 (25.7%) 1 (1.4%) 2 (2.9%)	1.00 0.39 0.84 0.17 0.65
Smoking	51 (72.9%)	48 (68.6%)	0.58
GFR mL/min/1.73 m^2^ (Me [25; 75])	83 [69; 89]	77 [65; 89]	0.46
LVEF, % (Me [25; 75])	60 [56; 66]	61 [57; 65]	0.74
CPB time, min (Me [25; 75])	80.0 [65; 100]	88 [75; 105]	0.05
Number of distal anastomoses performed (Me [25; 75])	3 [2; 3]	3 [3; 3]	0.13

NB: BMI—body mass index, CCS—Canadian Cardiovascular Society, CPB—cardiopulmonary bypass, GFR—glomerular filtration rate, LVEF—left ventricular ejection fraction, MI—myocardial infarction.

**Table 2 biomedicines-13-01999-t002:** Postoperative data.

Variables	Group 1 (*n* = 70)	Group 2 (*n* = 70)	*p*-Value
POAF, *n* (%)	9 (12.9%)	2 (2.9%)	0.03
Operative mortality, *n* (%)	0 (0%)	0 (0%)	-
Stroke, *n* (%)	0 (0%)	0 (0%)	-
Myocardial infarction, *n* (%)	1 (1.4%)	1 (1.4%)	1.00
Postoperative bleeding, *n* (%)	3 (4.3%)	1 (1.4%)	0.31
Number of patients with hydrothorax > 300 mL requiring thoracentesis, *n* (%)	23 (32.9%)	20 (28.6%)	0.58
WBC count in CBC (Me [25; 75]): Before surgery Day 1 post-surgery Day 3 post-surgery Day 5 post-surgery Day 10 post-surgery	7.2 × 10^9^/L [6.4; 7.8] 10.2 × 10^9^/L [8.8; 11.5] 9.5 × 10^9^/L [7.9; 10.9] 7.5 × 10^9^/L [6.5; 9.6] 8.6 × 10^9^/L [6.7; 9.9]	7.6 × 10^9^/L [6.5; 8.8] 11.4 × 10^9^/L [9.6; 13.0] 10.1 × 10^9^/L [8.7; 12.1] 8.8 × 10^9^/L [7.3; 10.0] 8.9 × 10^9^/L [7.5; 11.3]	0.08 0.01 0.11 0.07 0.16
CRP, mg/L (Me [25; 75]): Before surgery Day 1 post-surgery Day 3 post-surgery Day 5 post-surgery Day 10 post-surgery	4.0 [3.2; 6.0] 51.5 [28; 69] 126 [89,2; 166,0] 56 [31; 87,1] 21.9 [10.7; 37.0]	3.5 [2.1; 6.0] 57 [47; 69] 144.1 [109.8; 174.1] 45.4 [25.3; 82.0] 13.2 [6.9; 29.9]	0.07 0.08 0.02 0.52 0.0
AST, U/L (Me [25; 75]): Before surgery Day 1 post-surgery Day 5 post-surgery Day 10 post-surgery	23 [41; 73] 51 [41; 73] 25 [18.5; 46] 22 [17; 28]	19.5 [16.4; 24] 51 [41; 63] 29 [22; 43] 22.2 [17; 32]	0.07 0.94 0.50 0.49
ALT, U/L (Me [25; 75]): Before surgery Day 1 post-surgery Day 5 post-surgery Day 10 post-surgery	25 [15; 35.4] 32 [22; 48] 21.7 [15; 40] 28 [17; 40]	19.5 [16; 27.5] 30 [21; 45] 27.7 [20; 36] 27 [19; 34]	0.10 0.55 0.14 0.83
Creatinine, µmol/L (Me [25; 75]): Before surgery Day 1 post-surgery Day 5 post-surgery Day 10 post-surgery	88 [76; 99] 90 [78; 102] 88 [76; 98] 88 [77; 104]	84.5 [71; 103] 90 [76; 110] 82.5 [71; 102.5] 84 [77; 99]	0.70 0.79 0.57 0.87
Pneumonia, *n* (%)	9 (12.9%)	5 (7.1%)	0.26
Mediastinitis, *n* (%)	0	0	-
Pericarditis, *n* (%)	1 (1.4%)	0 (0%)	0.32
Surgical debridement of the sternum, *n* (%)	2 (2.9%)	4 (5.7%)	0.40
Gastrointestinal complications (diarrhea, nausea, or vomiting), *n* (%)	1 (1.4%)	0	0.32

NB: ALT—alanine aminotransferase, AST—aspartate aminotransferase, CBC—complete blood count, CRP—C-reactive protein, POAF—postoperative atrial fibrillation, WBC—white blood cell, quantitative data are presented as median and quartiles (Me [25; 75]).

## Data Availability

The datasets used and/or analyzed during the current study are available from the corresponding author on reasonable request.

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
