# Peer review of "A Combined Approach to the Prevention of Postoperative Atrial Fibrillation in Cardiac Surgery"

_biomedicines, 2025, doi:10.3390/biomedicines13081999_

Round 1

Reviewer 1 Report (New Reviewer)

Comments and Suggestions for Authors

Congratulations to the authors for their manuscript dealing with one the most common issue in post operative patients that underwent cardiac surgery; below you can find my comments:

Authors should clearly describe the randomization procedure: was it generated using a computer algorithm or based on sealed opaque envelopes?

Authors should indicate whether the individuals assessing the outcomes were blinded to the participants' group allocation.

In their statistics, authors should incorporate logistic regression analysis for the primary endpoin to control for established risk factors. Moreover they should report the number needed to treat to contextualize the clinical relevance of the findings. In addition authors should present confidence intervals for the reported odds ratio to enhance the interpretability of the effect estimate.

The current discussion does not sufficiently explore the mechanistic basis of colchicine’s effect, such as its role in inhibiting the NLRP3 inflammasome or mitigating pericarditis-related triggers. Moreover authors are encouraged to include in their discussion the fundamental concept of atrial cardiomyopathy which so relevant in the determination of AF (DOI:10.23736/S2724-5683.25.06725-0). Posterior pericardiotomy remains a rarely implemented surgical technique; it would be valuable to elaborate on its practical applicability, potential contraindications, and supporting evidence from prior studies.

Author Response

Dear Reviewer,

We sincerely appreciate the time and effort you have dedicated to evaluating our manuscript. Your insightful comments and constructive feedback have been invaluable in strengthening our work.

In response, we have carefully revised the manuscript, incorporating the suggested modifications to the best of our ability. To facilitate the review process, we have used the “track-changes” mode and provided a detailed point-by-point response to your comments, which has been uploaded to the journal system.

We hope these revisions sufficiently address the concerns raised and meet your expectations.

Reviewer 2 Report (New Reviewer)

Comments and Suggestions for Authors

Dikova et al. present a prospective randomised trial on the use of Colchicine to prevent POAF.  I would like to congratulate the authors on a well designed study and interesting results.

-Titel: “Innovative” in the title is quite a stretch. There exist other RCTs and Meta analysis (doi: 10.1016/j.amjcard.2024.09.003) investigating the benefit of Colchicine to reduce POAF.

- The study is a prospective randomised trial which investigated 140 patients undergoing CABG and the intervention group underwent treatment with Colchicine. Did the patients receive other drugs which could affect POAF ? ß-Blockers , Amiodaron, other anti-inflammatory eg NSAIDs ?

- Introduction: In my opinion,  the introduction is quite long. Colchicine is a known drug, and the mechanism of action is not required to be mentioned again. Please shorten the introduction.

-Methods: Did all patients undergo posterior pericardiotomy? Is it standard to close the pericardium ?

- Was  there any evidence of retained blood syndrome despite posterior pericardiotomy ?

Author Response

Dear Reviewer,

We sincerely appreciate the time and effort you have dedicated to evaluating our manuscript. Your insightful comments and constructive feedback have been invaluable in strengthening our work.

In response, we have carefully revised the manuscript, incorporating the suggested modifications to the best of our ability. To facilitate the review process, we have used the “track-changes” mode and provided a detailed point-by-point response to your comments, which has been uploaded to the journal system.

We hope these revisions sufficiently address the concerns raised and meet your expectations.

Round 2

Reviewer 1 Report (New Reviewer)

Comments and Suggestions for Authors

Congratulations to the authors for the revised version of their manuscript 

This manuscript is a resubmission of an earlier submission. The following is a list of the peer review reports and author responses from that submission.

Round 1

Reviewer 1 Report

Comments and Suggestions for Authors

Comments:

Post-operative atrial fibrillation (POAF) is one of the most common complications of cardiac surgery, occurring in 20% to 40% of patients. POAF is associated with longer hospital stays and higher health care costs, and, despite being short lived, is associated with worse long-term clinical outcomes including stroke, heart failure, recurrent hospitalization, and death.  Despite the interesting and promising proof-of-concept results from many studies, a number of important questions remain about the safety and efficacy of different methods to prevent POAF in patients undergoing cardiac surgery.

Dr. Diakova and colleagues performed a single-center, prospective, randomized controlled trial on the Colchicine Use for the Prevention of Postoperative Atrial Fibrillation in Cardiac Surgery. In this study, 140 patients underwent on-pump CABG were randomized into 2 groups of 70 patients each. They concluded that perioperative colchicine administration combined with pericardial fenestration during CABG with CPB is associated with a reduced POAF incidence, good tolerability, and does not contribute to an increased incidence of infectious complications or impaired liver and renal function. . 

It is an interesting idea, and the authors should be congratulated for their efforts. However, this study does not bring any new to the table. What is more, the design of trial and expression is confusing and log-winded. 

Here is some major problems and questions:

  1. The expression of Introduction is too tedious, and suggested to refine the wording.
  2. The statement of groups in Materials and Methods is contradictory with Abstract. In abstract section, group 1 was control group and received standard postoperative care, Group 2 was intervention group and received colchicine with intraoperative pericardial fenestration(line 20-24). In Materials section (line 119-128), Group 1, additionally  “posterior pericardiotomy and isolated pericardial drainage were performed intraoperatively”(line 121-122), patients in Group 2 underwent pericardial drainage intraoperatively using an original technique(line 127-128). 
  3. The Randomized Controlled Trial published on Lancet (2021 Dec 4; 398(10316):2075-2083. doi: 10.1016/S0140-6736(21)02490-9. Epub 2021 Nov 14. Posterior left pericardiotomy for the prevention of atrial fibrillation after cardiac surgery: an adaptive, single-centre, single-blind, randomised, controlled trial)  have shown that posterior left pericardiotomy is highly effective in reducing the incidence of atrial fibrillation after surgery on the coronary arteries, aortic valve, or ascending aorta, or a combination of these without additional risk of postoperative complications. So, in the trial, combining colchicine and posterior left pericardiotomy together in intervention group is not enough to prove that colchicine is effective, unless posterior left pericardiotomy is also used in the control group.
  4. In the case of NSAIDs, current data fail to support their efficacy in POAF prevention. Moreover, perioperative administration of NSAIDs may be associated with some severe safety considerations (Nomani, H.; Mohammadpour, A.H.; Moallem, S.M.H.; Sahebkar, A. Anti-inflammatory drugs in the prevention of post-operative atrial fibrillation: a literature review. Inflammopharmacology 2020, 28(1), 111-129, https://doi.org/10.1007/s10787-019-00653-x.). Why still use diclofenac routinely in control group? 
  5. What’s the definition of postoperative atrial fibrillation in this study? 
  6. The secondary endpoints in this trial is not comprehensive enough. The operative mortality, postoperative major adverse events, such as stroke, myocardial infarction,  bleeding should be included.
  7. Form Line 193 should be included in Result section.
  8. The expression of Results and Discussion section is not appropriate and long-winded, must be revised.
  9. line 127-128, improper application of references: reference 10 is on the anti-inflammatory drugs in the prevention of POAF, not on the pericardial drainage.
  10. 10.The decimal places in the P value should be consistent.
  11. 11.The limitation of study should be analyzed.

Reviewer 2 Report

Comments and Suggestions for Authors

The main problem of this study is its flawed methodology. Although the authors planned it as a randomized controlled trial (RCT), the fundamental principles of this study design were violated.

In an RCT, only one variable (the one under investigation) should differ between the compared groups. In this study, the authors have two types of variables in the intervention group (Group 2): colchicine and fenestration using the original technique. In the control group (Group 1), they used diclofenac (why not placebo?) and posterior pericardiectomy. Consequently, the effect on the primary endpoint in Group 2 is influenced not only by the use of colchicine, but also by the original fenestration technique. This comparison is invalid. This methodological error prevents the results of the study from being evaluated from the perspective of assessing the effect of colchicine on POAF.

I am not even considering all the other remarks regarding the description of the work. For instance, a cause-and-effect relationship between the interventions and outcomes is not demonstrated. Although the authors mention calculating odds ratios, these are not presented in the article.
And much else...

Similar studies have been conducted on large patient populations. Furthermore, there are quite recent meta-analyses in this field.
For example:

https://academic.oup.com/europace/article/25/7/euad169/7204264

https://www.mdpi.com/2308-3425/9/10/363 

The novelty of this work is unclear.